# When Multimodal Models "Burn Out": Diagnosing and Healing Modality Fatigue via Detect–Adapt–Compensate

## Abstract

In long-context multimodal reasoning, models often begin to "burn out"—not because of architectural flaws, but because one or more input modalities gradually lose their expressiveness. *Is this merely an attention failure? Or is the modality itself fatigued?* We propose a new perspective on this degradation: **Modality Fatigue**, a phenomenon where the model's activation and responsiveness to certain modalities decay over time, manifesting as attention attenuation, fusion drift, semantic shift, and loss of task sensitivity. Unlike prior approaches that focus on modeling inter-modal attention patterns or equationment graphs, we shift the lens to the evolving internal state of each modality. We conceptualize modality fatigue as a dynamic decline in each modality's "vital sign," modeled through its activation signal trajectory. Concretely, we introduce the **Modality Activation Decay Detector (MAD)** detector to monitor each modality's instantaneous activation $\alpha_m(t)$ and its change rate $\delta_m(t)$, while dynamically computing a fatigue-triggering threshold $\tau_m(t)$ from historical trends. Once fatigue is detected, the **Modality Alternation & Compensation Controller (MAC)** adaptively adjusts the fusion path and recall compensation. It controls the integration of current perception and retrieved memory via a learnable gate $\lambda_m(t)$, thereby restoring under-utilized modality signals. Our method sidesteps the need for full attention matrices or inter-modal graph modeling. Instead, it decomposes modality state tracking into independent one-dimensional activation curves, enabling lightweight monitoring and fine-grained control with high interpretability. Across various long-context benchmarks, our framework demonstrates encouraging capabilities in preserving modality balance, enhancing fusion robustness, and mitigating information drift and omission. By uncovering and addressing modality fatigue through transparent, signal-based modeling, we take a step toward building multimodal systems that can perceive their own internal states and adapt accordingly.

## 1 Introduction

*Do multimodal models "burn out" over time?* Despite recent progress in multimodal large language models (MLLMs), we observe a puzzling phenomenon: as input length increases and reasoning deepens, models begin to lose touch with certain modalities, even when those modalities remain present and relevant. For instance, in visual question answering, models may rely heavily on the question text while progressively ignoring the image; in audio-captioning, linguistic outputs may remain fluent despite audio cues fading from memory. We term this phenomenon *modality fatigue*: a gradual and systematic decline in modality-specific activation over long-context reasoning. As illustrated in Figure 1, modality traces exhibit diminishing signal strength, causing the model to converge toward default or biased behaviors (*e.g., language dominance*). Crucially, this is not due to corrupted inputs or noisy supervision-all modalities are available and properly equationed. The problem may lie in how the model manages internal attention, fusion, and memory usage over time. Unlike existing studies that focus on input-level modality balancing or static fusion, we take a process-centric view: modality degradation is an evolving state, not a static mismatch. If models fail to sense when a modality fades, they cannot recover its influence or adapt their reasoning accordingly. Addressing this subtle yet pervasive fatigue requires a new perspective—one that moves beyond attention scores

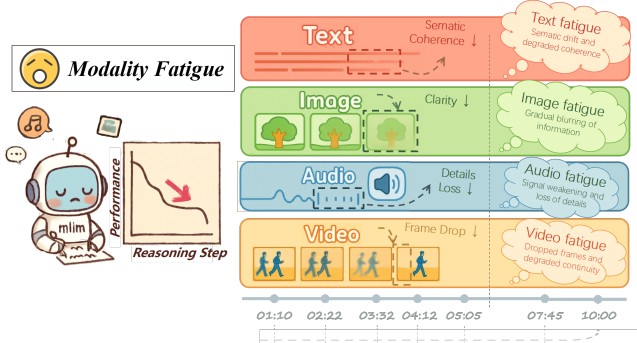

Figure 1: **Modality fatigue in long-context reasoning.** As multimodal models process longer sequences, individual modalities (e.g., vision, audio, Text, Image) exhibit activation decay, gradually losing their content and influence on the final prediction. Our proposed framework tracks these signals over time and compensates for fading paths.

and toward dynamic signals of modal activation, as explored in dynamic fusion frameworks that adapt to instance-wise modality relevance Xue & Marculescu (2023), condition-aware reliability estimation Brödermann et al. (2024), and continuous modality-domain attention for degrading or missing modalities Fan et al. (2024).

*Why do current multimodal models fail to notice this fatigue?* A closer inspection reveals that most existing architectures are not designed to monitor the evolving "health" of each modality throughout the reasoning process. While attention mechanisms allow modalities to interact at each step, they offer no explicit signal for how a modality's contribution changes over time. Once initial attention weights are set: often biased toward text and there is no feedback loop to detect whether other modalities are fading, weakening, or becoming sidelined. Importantly, modality fatigue is *not* a problem of corrupted input, low-resolution signals, or insufficient equationment at the fusion layer. Even when all inputs are intact and semantically relevant, the model may still progressively abandon useful modalities simply because it lacks introspective awareness of how modal activations evolve. This disconnect leads to over-reliance on dominant modalities and a loss of grounding, especially in tasks requiring cross-modal integration or temporal continuity. We argue that true robustness in MLLMs requires more than static fusion or per-layer attention balancing. It demands a new level of internal observability: the ability to track fine-grained signals of modality engagement—what we call the modality's "vital signs." These signals should capture both the strength and dynamics of activation across steps, enabling the model to sense early signs of fatigue and proactively intervene.

To address **modality fatigue** as a process of gradual degradation rather than explicit failure, we adopt a **micro-level perspective** that tracks each modality's evolving contribution through lightweight and interpretable **activation signals**. For each modality $m$ at time step $t$, we define three signals: (1) **Activation Level** $\alpha_m(t)$, which measures the aggregated magnitude of its feature tokens and reflects the overall contribution of modality $m$; (2) **Activation Change** $\delta_m(t)$, which captures the rate at which the modality's activation increases or decreases, indicating the onset of fatigue; and (3) a dynamic **Fatigue Threshold** $\tau_m(t)$, which is derived from the recent trajectory of $\delta_m(t)$ and identifies statistically significant drops in engagement. These signals provide a fine-grained, real-time view of modality health that conventional attention maps often fail to capture. In addition to per-modality monitoring, we compute global descriptors such as the mean and variance of $\alpha_m(t)$ across all modalities, which serve as system-level indicators of activation imbalance. These summaries help contextualize the decline of individual modalities within the broader system dynamics, offering insights into interaction asymmetries, modality overreliance, or unstable fusion behavior. Importantly, this formulation is modality-agnostic and does not require any architecture-specific modification, allowing straightforward application across vision, language, and audio modalities.

To operationalize modality fatigue detection and response, we introduce the **Modality Activation Decay Detector (MAD)** and **Modality Alternation & Compensation Controller (MAC)**: a two-stage self-regulation framework for real-time fatigue awareness and correction. **MAD** continuously tracks the vital signs defined in the previous section. When a modality's activation declines below its adaptive threshold (i.e., $\delta_m(t) < -\tau_m(t)$), MAD flags it as entering a *fatigue state* and raises an

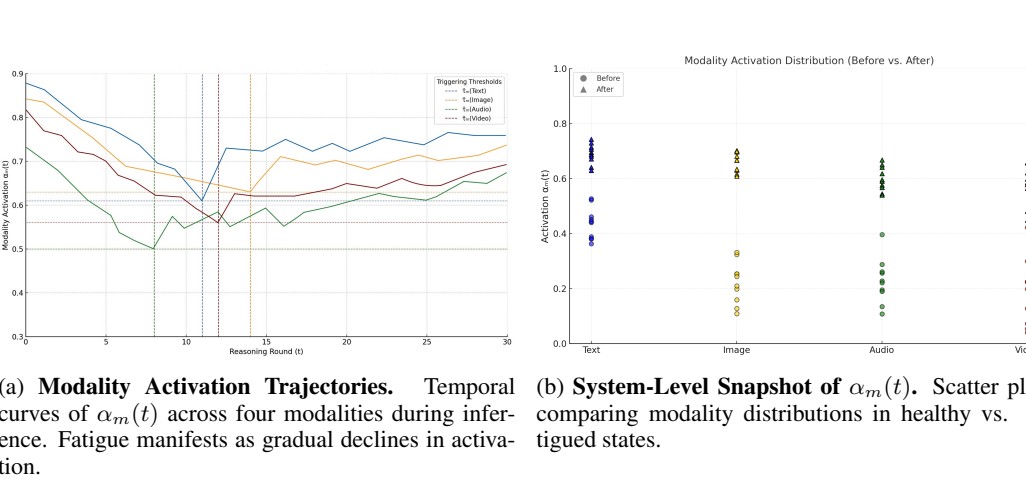

(a) **Modality Activation Trajectories.** Temporal curves of $\alpha_m(t)$ across four modalities during inference. Fatigue manifests as gradual declines in activation.

(b) **System-Level Snapshot of $\alpha_m(t)$.** Scatter plots comparing modality distributions in healthy vs. fatigued states.

Figure 2: Side-by-side comparison of modality activation dynamics.

alert to reconfigure the fusion behavior. This step ensures early and precise detection, preventing long-term modality collapse. **MAC** then takes over to re-route and stabilize the fusion path. If a modality is fatigued, MAC dynamically retrieves salient past signals from memory to restore its representation, rather than discarding or freezing it. Conversely, if the modality remains healthy, MAC amplifies its current signal to maintain clarity. The result is a context-aware fusion flow that adapts to modality health over time. This modular design enables a powerful "detect–adapt–restore" pipeline. For example, in a multimodal dialogue task, if the visual stream fades due to occlusion or temporal decay, MAD identifies the drop, and MAC recalls earlier vision features equationed with the current query, blending them with present cues via a soft compensation gate. This allows the model to maintain visual grounding, even when raw inputs weaken. Together, MAD and MAC act as a lightweight, plug-and-play module that enables multimodal models to self-monitor their reasoning health, react adaptively, and maintain stable, modality-aware fusion paths.

Extensive experiments on long-context multimodal tasks demonstrate that our framework improves robustness, maintains semantic fidelity, and enhances reasoning stability, without introducing architectural burden or requiring external supervision. In summary, this work takes a step toward understanding and mitigating modality fatigue: a previously underexplored phenomenon in long-context multimodal reasoning. Rather than relying on attention-based fusion graphs or architectural overhauls, we advocate a micro-level control perspective based on per-modality signal tracking. Our contributions are three folds:

- We propose a novel formulation of **modality fatigue**, modeled as the temporal decay of per-modality activation trajectories. We define interpretable real-time indicators, including activation level $\alpha_m(t)$, its derivative $\delta_m(t)$, and the adaptive threshold $\tau_m(t)$, to identify and track fatigue states at the signal level.

- We design a unified control mechanism combining the **Modality Activation Decay Detector (MAD)** and the **Modality Alternation & Compensation Controller (MAC)**. This two-stage process enables online fatigue diagnosis and dynamic fusion reconfiguration by: (i) detecting early-stage modality degradation; and (ii) recalling historical signals and adaptively adjusting fusion weights, ensuring reasoning resilience even under degraded inputs.

- Our modules are plug-and-play architecture that require no additional supervision, and integrates seamlessly with existing MLLMs. They generalizes across diverse long-context multimodal tasks, offering a practical solution for fatigue-aware reasoning at scale.

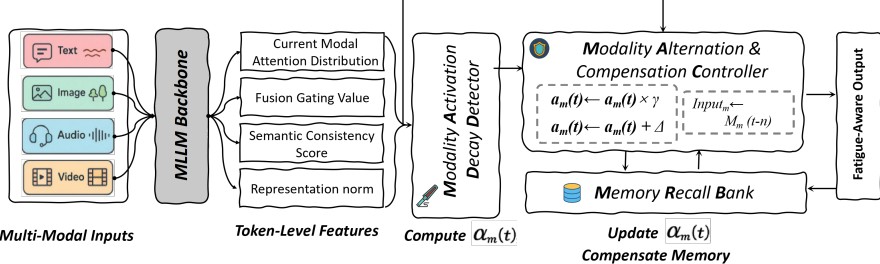

Figure 3: **Signal Extraction, Fatigue Detection, and Fusion Compensation.** The system extracts token-level signals to compute activation $\alpha_m(t)$. The MAD module detects fatigue ($s_m(t)$), and the MAC controller compensates via gated recall from memory using $\lambda_m(t)$.

## 2  WHAT IS MODALITY FATIGUE? MODELING WITH MODALITY ACTIVATION SCORE

Multimodal reasoning relies on integrating diverse inputs: language, vision, and more. Yet, in long or complex tasks, this coordination often degrades unevenly. We define this as **modality fatigue**: a progressive decline in a model's responsiveness to specific modalities during multi-step inference.

**Key Symptoms.** Modality fatigue reflects deeper internal imbalances, not just performance loss. It manifests through: (1) **Attention Attenuation** — declining focus on modality-specific features; (2) **Fusion Instability** — inconsistent gating and weighting; (3) **Semantic Drift** — output diverging from modality input; (4) **Task Insensitivity** — failure to detect modality relevance for the task.

**Modeling Fatigue.** We track fatigue using the *Modality Activation Score* $\alpha_m(t)$ and its dynamics:

- $\alpha_m(t)$: Current activation level of modality $m$;
- $\delta_m(t)$: Temporal change in activation;
- $\tau_m(t)$: Fatigue threshold;
- $s_m(t)$: Trigger signal when decline exceeds threshold.

As shown in Figure 2a, all modalities exhibit declining $\alpha_m(t)$, especially image and video, indicating susceptibility to temporal fatigue. Thresholds $\tau_m(t)$ help identify when intervention is needed.

**Global Activation Landscape.** Beyond per-modality trends, we analyze the distribution of $\alpha_m(t)$ to capture system-wide reasoning states. As shown in Figure 2b, fatigued states exhibit both lower mean activation and reduced variance, indicating two failure modes: (1) **Low Mean** — overall inattentiveness; (2) **Low Variance** — loss of modality distinction.

Together, the temporal evolution and cross-modal dynamics of $\alpha_m(t)$ provide an interpretable and actionable lens to characterize modality fatigue.

## 3  DIAGNOSING AND HEALING MODALITY FATIGUE

We introduce a unified diagnostic and healing framework that dynamically detects fatigued modalities and restores multimodal equilibrium through adaptive intervention. Figure 4 presents an overview of our proposed pipeline.

### 3.1  DIAGNOSING FATIGUE VIA MODALITY ACTIVATION DECAY DETECTOR (MAD)

To diagnose modality fatigue at a fine-grained level, we introduce **Modality Activation Decay Detector (MADD)**, a lightweight signal tracking module that continuously monitors the dynamic usage and degradation of each modality during reasoning. As shown in Figure 4 (step 2) and detailed in Figure 3, MADD computes a per-step fatigue signal by tracing the activation status of each modality over time.

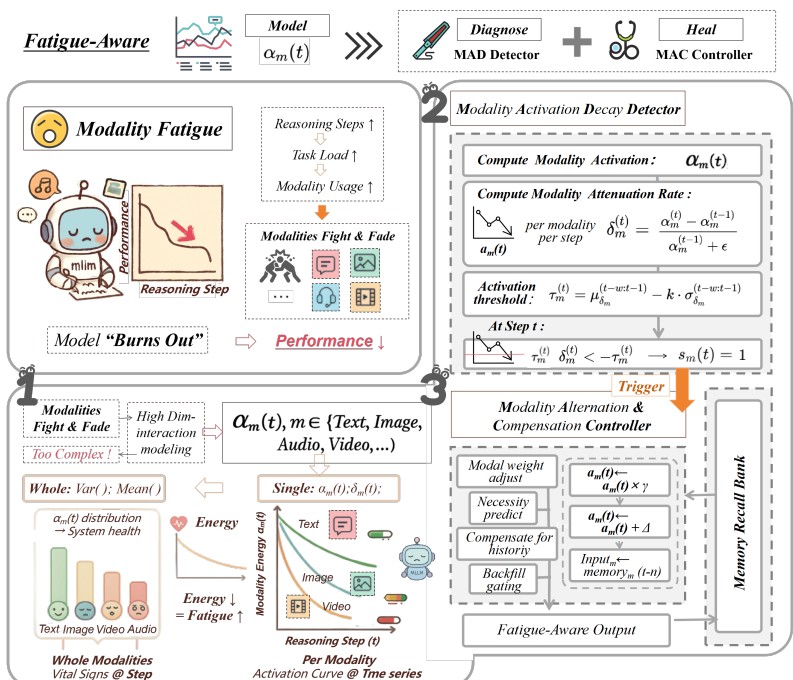

Figure 4: **Workflow of Diagnosing and Healing Modality Fatigue.** Our method proceeds in three stages: (1) *Degradation Detection* based on output collapse or weakened modality cues; (2) *MAD Module* computes per-modality activation $\alpha_m(t)$ and decay $\delta_m(t)$, triggering fatigue signal $s_m(t)$ via thresholding with $\tau_m(t)$; (3) *MAC Controller* compensates fatigued modalities via memory recall and gated fusion to produce fatigue-aware outputs. Arrows and equations (e.g., $\alpha_m(t) \leftarrow \alpha_m(t) \cdot \gamma$) are symbolic representations of key updates.

**Modality Activation Signal.** We define $\alpha_m(t)$ as the attention-weighted L2 activation mean of modality $m$ at timestep $t$. This formulation captures not only the representational strength of each token but also the model's focus on them, offering a faithful measure of actual modality usage:

$$\alpha_m(t) = \sum_{i=1}^{N_m} a_i^{(m,t)} \, \|\mathbf{h}_i^{(m,t)}\|_2 \tag{1}$$

Here, $a_i^{(m,t)}$ is the attention weight and $\mathbf{h}_i^{(m,t)}$ is the hidden representation of the $i$-th token in modality $m$. $\alpha_m(t)$ thereby serves as a micro-level vital sign for each modality, enabling localized tracking across time and context.

This signal departs from global heuristics like average attention or representation norm—those may miss subtle per-modality variations. In contrast, $\alpha_m(t)$ enables more precise and explainable reasoning by focusing on actual signal strength and usage per modality.

**Per-step Attenuation and Adaptive Threshold.** While $\alpha_m(t)$ reflects the current activation level, detecting fatigue requires observing how this activation evolves. We thus compute a per-step decay rate $\delta_m(t)$ as:

$$\delta_m(t) = \frac{\alpha_m(t) - \alpha_m(t-1)}{\alpha_m(t-1) + \varepsilon} \tag{2}$$

This normalized change rate reveals whether a modality is losing vitality across reasoning steps. However, to avoid transient fluctuations, we introduce an adaptive threshold $\tau_m(t)$ via a sliding window over recent $\delta_m$ values:

$$\tau_m(t) = \mu_{\delta_m}^{[t-w:t-1]} - \sigma_{\delta_m}^{[t-w:t-1]} \tag{3}$$

where $w$ is the window size and $\mu$, $\sigma$ are the mean and standard deviation. A sharp negative drop below this threshold indicates fatigue onset.

**Fatigue Signal Trigger.** We then define the binary fatigue flag $s_m(t)$ that triggers compensation when a modality's decay exceeds the adaptive threshold:

$$s_m(t) = \begin{cases} 1, & \delta_m(t) < -\tau_m(t) \\ 0, & \text{otherwise} \end{cases} \tag{4}$$

This signal is lightweight to compute, requires no supervision, and enables online detection of modality-specific degradation.

### 3.2 HEALING VIA MODALITY ALTERNATION & COMPENSATION CONTROLLER (MACC)

Upon detecting modality fatigue $s_m(t) = 1$ signal sent by MAD, we activate the **Modality Alternation & Compensation Controller (MACC)** to restore the degraded modality through two functional branches: *Modal weight adjust* and *Compensate for history* are shown in Workflow 3.

**Step 1: Activation Reweighting for Non-Fatigued Modalities.** For *all modalities*, MACC first quantifies their contextual relevance to the current reasoning query via cosine similarity:

$$r_m(t) = \cos\left(\mathbf{q}(t), \mathbf{f}_m(t)\right) \tag{5}$$

which quantifies alignment between the current query $\mathbf{q}(t)$ and modality feature $\mathbf{f}_m(t)$. Based on this, we update the effective activation:

$$\tilde{\alpha}_m(t) = \alpha_m(t) + (1 - s_m(t)) \cdot r_m(t) - s_m(t)(1 - \gamma)\alpha_m(t) \tag{6}$$

This simultaneously enhances relevant and healthy modalities, while softly decaying fatigued and irrelevant ones.

**Step 2: Memory Compensation for Fatigued Modalities.** For fatigued modalities, we attempt to recover prior memory to reconstruct a trustworthy signal. First, we retrieve memory:

$$\mathbf{M}_m(t) = \sum_{i=1}^{t-1} \rho_m^{(i)}(t) \cdot \mathbf{f}_m(i) \tag{7}$$

where the retrieval weights are defined as:

$$\rho_m^{(i)}(t) = \frac{\exp\left(\mathbf{q}(t)^\top \mathbf{f}_m(i)\right)}{\sum_{j=1}^{t-1} \exp\left(\mathbf{q}(t)^\top \mathbf{f}_m(j)\right)} \tag{8}$$

We then compute the semantic discrepancy:

$$\delta_{\text{sim}}(t) = \text{sim}(\mathbf{f}_m(t), \mathbf{q}(t)) - \text{sim}(\mathbf{M}_m(t), \mathbf{q}(t)) \tag{9}$$

This difference determines a confidence gate:

$$\lambda_m(t) = \frac{1}{1 + \exp\left(-k \cdot \delta_{\text{sim}}(t)\right)} \tag{10}$$

Finally, we construct the compensated representation:

$$\hat{\mathbf{f}}_m(t) = \lambda_m(t) \cdot \mathbf{f}_m(t) + (1 - \lambda_m(t)) \cdot \mathbf{M}_m(t) \tag{11}$$

This ensures that degraded modalities can selectively reactivate prior knowledge while preserving current input when reliable.

After all modality representations $\hat{\mathbf{f}}_m(t)$ are updated, we perform the final fatigue-aware fusion.

### 3.3 FUSION WITH FATIGUE-AWARE MODALITY WEIGHTS

To generate the overall multimodal representation $\mathbf{z}(t)$, we perform soft attention over the adjusted activations:

$$w_m(t) = \frac{\exp\left(\alpha_m^{(t)} \cdot r_m^{(t)}/T\right)}{\sum_j \exp\left(\alpha_j^{(t)} \cdot r_j^{(t)}/T\right)} \tag{12}$$

Table 1: Forgetting Ratio (%) of each task in Order A and Order B. Order A: Flickr30k → Audio-Caps → MSVD-QA → OK-VQA → Clotho-AQA → MSR-VTT. Order B: Flickr30k → OK-VQA → AudioCaps → Clotho-AQA → MSR-VTT → MSVD-QA. 0.00 indicates last-task; negative denotes positive transfer. Bold: best, underline: second best.

| | | | | Order A | | | | |
|---|---|---|---|---|---|---|---|---|
| Method | Flickr30k | AudioCaps | MSVD-QA | OK-VQA | Clotho-AQA | MSR-VTT | Avg Acc ↑ | Avg Forget ↓ |
| FT | 84.21 | 70.68 | 58.57 | 6.06 | 9.64 | 0.00 | 17.90 | 38.19 |
| LoRA | 71.02 | 64.15 | 65.45 | 4.12 | 2.23 | 0.00 | 24.35 | 34.50 |
| MOE-LoRA | 63.14 | 57.89 | 60.83 | 3.44 | 5.12 | 0.00 | 28.10 | 31.74 |
| EWC | 55.93 | 50.44 | 55.69 | 14.24 | 10.93 | 0.00 | 36.02 | 31.21 |
| PGP | 25.13 | 43.98 | 62.17 | 1.08 | 3.45 | 0.00 | 39.95 | 22.64 |
| CL-MOE | 30.66 | 58.12 | 36.58 | 3.12 | 0.87 | 0.00 | 43.62 | 21.56 |
| **Ours** | **9.07** | **25.68** | **26.44** | **-7.15** | **-0.05** | 0.00 | **49.05** | **9.00** |

| | | | | Order B | | | | |
|---|---|---|---|---|---|---|---|---|
| Method | Flickr30k | OK-VQA | AudioCaps | Clotho-AQA | MSR-VTT | MSVD-QA | Avg Acc ↑ | Avg Forget ↓ |
| FT | 95.23 | 72.35 | 68.44 | 51.48 | 48.12 | 0.00 | 16.95 | 60.57 |
| LoRA | 91.02 | 68.44 | 68.44 | 42.05 | 39.00 | 0.00 | 25.35 | 54.61 |
| MOE-LoRA | 78.35 | 59.18 | 59.18 | 29.76 | 28.00 | 0.00 | 31.05 | 46.46 |
| EWC | 92.27 | 68.87 | 68.87 | 36.45 | 35.00 | 0.00 | 37.85 | 53.85 |
| PGP | 53.27 | 24.35 | 24.35 | 13.52 | 6.12 | 0.00 | 45.30 | 22.58 |
| CL-MOE | 38.86 | 31.63 | 31.63 | 11.42 | 5.50 | 0.00 | 51.50 | 21.42 |
| **Ours** | **14.57** | **3.12** | **14.26** | **2.00** | **0.00** | 14.26 | **49.80** | **8.64** |

These weights prioritize healthy, semantically relevant modalities while demoting fatigued or unreliable ones. The final representation is:

$$\mathbf{z}(t) = \sum_m w_m(t) \cdot \hat{\mathbf{f}}_m(t) \tag{13}$$

This two-branch strategy (adjust + compensate) jointly stabilizes long-horizon multimodal reasoning by continuously restoring modality fidelity.

# 4 EXPERIMENTS

## 4.1 EXPERIMENTAL SETUP

**Datasets.** We evaluate our method on six multimodal reasoning tasks that span static and temporal modalities across diverse domains: Image Captioning (Flickr30K Young et al. (2014)), Audio Captioning (AudioCaps Kim et al. (2019)), Video Captioning (MSR-VTT Xu et al. (2016)), Image QA (OK-VQA Marino et al. (2019)), Audio QA (Clotho-AQA Lipping et al. (2022)), and Video QA (MSVD-QA Xu et al. (2017)). Each round presents a distinct task in a fixed order (Order A or Order B). Additional dataset statistics are provided in the Appendix.

**Baselines.** We compare against six representative baselines: fine-tuning (FT) Howard & Ruder (2018), LoRA Hu et al. (2021), MoELoRA Luo et al. (2024), Elastic Weight Consolidation (EWC) Kirkpatrick et al. (2017), Progressive Prompts (PGP) Razdaibiedina et al. (2023), and Continual Learning MoE (CL-MoE) Huai et al. (2025). All methods use the same frozen multimodal encoders and language backbone. Our method integrates two additional modules, MAD and MAC, for detecting and mitigating modality fatigue during inference.

**Evaluation Metrics.** For task performance, we use CIDEr Vedantam et al. (2015) for captioning and answer accuracy for QA tasks, following prior work Panagopoulou et al. (2023). To assess fatigue and recovery behavior, we adopt a two-level evaluation: (1) degradation signals, including Fusion Bias, Entropy Change, Faithfulness, and Average Forgetting; (2) control behavior metrics, including Fatigue Trigger Rate (FTR), Recovery Gain, and Compensation Usage Rate. Full metric definitions are available in Appendix.

Table 2: Fatigue-Process Metrics across methods (Order A).

| Method | FTR ↓ | Fusion-Bias ↓ | Entropy Δ ↓ | Avg Forget ↓ |
|---|---|---|---|---|
| FT | 0.33 | 0.25 | 0.36 | 38.19 |
| LoRA | 0.29 | 0.23 | 0.34 | 34.50 |
| MOE-LoRA | 0.27 | 0.21 | 0.32 | 31.74 |
| EWC | 0.28 | 0.22 | 0.31 | 31.21 |
| PGP | 0.22 | 0.16 | 0.26 | 22.64 |
| CL-MOE | 0.20 | 0.17 | 0.24 | 21.56 |
| **Ours** | **0.11** | **0.08** | **0.12** | **9.00** |

## 4.2 TASK-LEVEL FATIGUE AND END PERFORMANCE

We begin by investigating whether modality fatigue leads to task-level degradation across long-context sequences, and whether our controller (MAD and MAC) improves final performance. We consider two task orders that alternate modalities and gradually increase cognitive load:

- **Order A:** Image Captioning $\rightarrow$ Audio Captioning $\rightarrow$ Video QA $\rightarrow$ Image QA $\rightarrow$ Audio QA $\rightarrow$ Video Captioning
- **Order B:** Image Captioning $\rightarrow$ Image QA $\rightarrow$ Audio Captioning $\rightarrow$ Audio QA $\rightarrow$ Video Captioning $\rightarrow$ Video QA

**Overall Fatigue Trend.** Table 1 reports the forgetting ratio for each task. Higher values reflect stronger degradation from early to later rounds. Baselines such as FT, LoRA, and EWC show substantial forgetting, especially on early tasks like `Flickr30k` and `AudioCaps`. For instance, FT forgets 84.21% on `Flickr30k` in Order A and 95.23% in Order B.

**Our Method's Improvement.** Our method yields the lowest average forgetting: 9.00% in Order A and 8.64% in Order B, clearly outperforming the strongest baseline CL-MOE (25.68%, 21.42%). In some cases, our controller even improves performance over time, such as $-7.15\%$ forgetting on `OK-VQA` in Order A, suggesting effective transfer across modalities.

Table 3: Trigger–Recovery Trace on Image-Captioning (Order A). Trigger condition: $H_{\text{img}} > 2.30$ and forget $\geq 15\%$. MAC window $K = 2$.

| Round | Avg Forget ↓ | $H_{\text{img}}$ | Fusion-Bias | Trigger | MAC |
|---|---|---|---|---|---|
| Start (0) | 0.00 | 1.28 | 0.02 | 0 | Off |
| Pre-Trig (11) | 6.20 | 1.94 | 0.15 | 0 | Off |
| Trigger (12) | 15.10 | 2.48 | 0.24 | 1 | On |
| +1 (13) | 10.30 | 1.72 | 0.08 | 0 | On |
| Final (30) | 9.00 | 1.78 | 0.09 | 0 | Off |

**Summary.** These findings confirm that modality fatigue arises even in semantically coherent sequences. Simple fine-tuning fails to retain early-task competence, while our controller adaptively regulates attention and memory to preserve performance throughout. Full metric breakdowns are available in Appendix.

### 4.2.1 CONTROLLER EFFECTIVENESS: DO MAC AND MAD ENABLE FATIGUE RECOVERY?

We assess the effectiveness of our controller modules MAD and MAC in detecting and mitigating modality fatigue. As shown in Table 2, in a 30-round Image Captioning trace (Order A), MAC is triggered at round 12 when image entropy exceeds 2.30 and forgetting surpasses 15%. Forgetting drops to 10.30% in the next round and stabilizes at 9.00%, with concurrent reductions in entropy and fusion bias indicating improved modality balance. our method achieves the lowest forgetting (9.00%) and FTR (0.11), along with minimal fusion bias (0.08) and entropy shift (0.12). It is also the only method that shows a positive recovery gain ($+0.14$) and nonzero compensation usage (0.007), reflecting effective regulation. Ablation results (Table. 4) show that removing MAD raises forgetting to

Table 4: **Ablation study showing the effect of removing MAC or MAD.** Removing MAC results in no recovery gain, validating its central role.

| Variant | Avg Forget ↓ | FTR ↓ | Recovery Gain ↑ | Comp Usage ↑ |
|---|---|---|---|---|
| Full | **9.0** | **0.11** | **0.14** | **0.22** |
| w/o MAD | 19.8 | 0.28 | 0.06 | 0.00 |
| w/o MAC | 15.9 | 0.11 | 0.00 | 0.00 |

19.8% and FTR to 0.28, while removing MAC removes recovery gain entirely. Compensation usage drops to zero in both cases, confirming that MAD enables early detection and MAC is essential for correction. Together, these results demonstrate that the two modules form a reliable and interpretable controller for managing fatigue in multimodal reasoning.

### 4.2.2 COMPENSATION ANALYSIS: HOW DOES THE CONTROLLER REACT AND ADAPT?

To better understand fatigue dynamics, we visualize two aspects: (1) the temporal evolution of fatigue indicators, and (2) modality transition patterns before and after fatigue. These reveal how our controller detects fatigue and initiates compensation. Figure 5(left) shows forgetting and image entropy across 30 dialogue rounds. At Round 12, both metrics peak, triggering the MAC controller. Post-intervention, forgetting and entropy drop, indicating recovery. Figure 5(right) tracks fusion

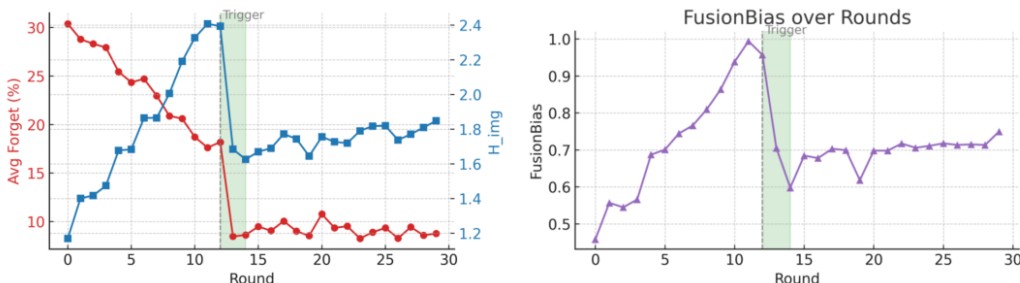

Figure 5: **Fatigue Trigger and Recovery Dynamics.** Forgetting rate (red), image entropy $H_{\text{img}}$ (blue), and fusion bias (purple) over 30 rounds. A trigger at `Round 12` (dashed line) activates the controller (green area), leading to recovery in performance and modality balance.

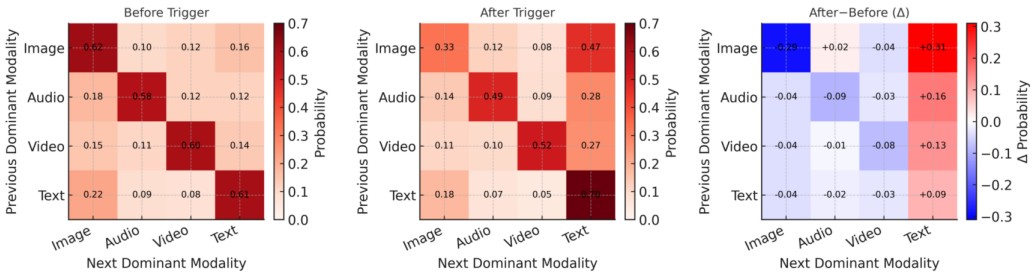

Figure 6: **Modality Transition Structure.** Heatmaps of dominant modality transitions before (Rounds 0–11) and after (Rounds 12–30) fatigue. Post-fatigue, shifts toward the text modality reveal modality escape under stress.

bias, which rises before fatigue due to growing modality imbalance and decreases after intervention, showing restored fusion stability. Figure 6 presents modality transition heatmaps. Before Round 12, dominant modalities show stable self-loops. After fatigue, transitions shift toward text dominance, reducing visual and audio persistence. The differential map confirms a clear flow toward text, reflecting "modality escape." Our controller detects this shift and compensates accordingly. Together, these visualizations validate the behavioral logic of fatigue detection and recovery. The combined signals of forgetting, entropy, and bias serve as effective triggers, enabling interpretable and adaptive fatigue mitigation.

## 5 CONCLUSION

We introduce modality fatigue as a progressive decline in modality-specific activation that undermines long-context multimodal reasoning. To detect and mitigate this subtle yet widespread issue, we propose a lightweight and interpretable control framework consisting of MAD for real-time signal monitoring and MAC for adaptive fusion reconfiguration. Our approach models each modality's activation trajectory as a dynamic vital sign, enabling timely diagnosis and targeted recovery. Through extensive experiments across six benchmarks and two task orders, our method consistently achieves the lowest forgetting rates, restores degraded modalities, and improves semantic and fusion stability. These results underscore the value of internal signal awareness in building more resilient, adaptive, and self-regulating multimodal systems.

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

## A  Extended Modeling of Modality Fatigue

This section expands upon the fatigue modeling framework in Section 3. We provide deeper analysis of the four core symptoms of modality fatigue and explain how they emerge from measurable signal patterns. These symptoms include: *attention attenuation*, *fusion instability*, *semantic drift*, and *task insensitivity*. Each reflects a different failure mode in multimodal reasoning, yet all can be tracked through the evolution of activation signals.

### A.1  Key Symptoms Revisited: Visual Traces of Fatigue

**Attention Attenuation.**  This symptom refers to the model's declining ability to focus on useful features within a specific modality. A consistent drop in activation level $\alpha_m(t)$, especially when combined with a sharply negative change rate $\delta_m(t)$, indicates that the modality is being ignored even when relevant. This often happens in visual or audio channels during long sequences. As attention weakens, the model compresses semantic diversity, and output quality drops accordingly.

**Fusion Instability.**  Fusion instability occurs when the model starts blending modalities unevenly. As some modalities degrade, the model shifts weight to others in an unbalanced way. We observe this in the increasing variance of fusion weights, which correlates with fatigue in weaker streams. This imbalance disrupts the expected integration behavior. Stabilization typically only occurs when the MAC controller is triggered to reconfigure the fusion strategy.

**Semantic Drift.**  Semantic drift happens when the model's output detaches from the intended modality. In vision-language tasks, for example, a fatigued visual stream leads the model to default to text-based answers, regardless of the image. This drift is usually preceded by a steep drop in $\alpha_{\text{vision}}(t)$ and a rise in output entropy $\Delta H$, signaling confusion. In Table 1, this pattern aligns with spikes in forgetting on visual tasks, confirming the behavioral impact.

**Task Insensitivity.**  Task insensitivity describes the model's failure to shift modality focus as task demands change. For instance, when switching from an audio captioning task to a visual question-answering task, the model may continue relying on the audio stream. This rigid behavior is reflected in low inter-modal variance in $\alpha_m(t)$, meaning all modalities are treated similarly despite different importance. As a result, the model misses key signals and loses contextual precision.

These four symptoms are not isolated issues. They reflect deeper breakdowns in how the model tracks and integrates modality-specific information. By monitoring $\alpha_m(t)$ and $\delta_m(t)$ over time, we gain interpretable and actionable signals to detect such failures early. Compared to static attention scores, these activation signals offer a clearer view of how the model is reasoning. They form the basis of our fatigue-aware control system, which responds to these failures in real time.

### A.2  Modeling Activation Dynamics in Complex Tasks

Modality fatigue does not occur all at once. It is a gradual and task-dependent process, shaped by how each modality's activation changes over time and how responsive it remains to the task context. In this section, we analyze two key signals: the modality attenuation rate $\delta_m(t)$ and the adaptive threshold $\tau_m(t)$. These signals enable accurate and interpretable detection of fatigue during long or multi-stage reasoning.

**Modality Attenuation Rate $\delta_m(t)$.**  The signal $\delta_m(t)$ quantifies how quickly a modality's activation is increasing or decreasing between steps. While $\alpha_m(t)$ tells us how strongly the model is using modality $m$ at a specific moment, $\delta_m(t)$ shows whether this usage is rising or falling. In a well-functioning system, healthy modalities show fluctuating but stable or slightly positive $\delta_m(t)$ values, indicating continued engagement. In contrast, modalities undergoing fatigue exhibit a consistent

downward trend, where $\delta_m(t)$ remains negative over time. This gradual loss of influence often leads to performance drops, especially when the affected modality is essential for the task at hand.

**Adaptive Threshold** $\tau_m(t)$. To prevent reacting to short-term noise, we define a threshold $\tau_m(t)$ that adapts over time. It is computed using a sliding window of past $\delta_m(t)$ values and serves as a statistical baseline for identifying abnormal decline. Specifically, we define:

$$\tau_m(t) = \mu_{\delta_m}^{[t-w:t-1]} - \sigma_{\delta_m}^{[t-w:t-1]} \tag{14}$$

Here, $\mu_{\delta_m}$ and $\sigma_{\delta_m}$ represent the mean and standard deviation of recent $\delta_m$ values, and $w$ is the window size. When $\delta_m(t)$ drops below this threshold, we treat it as a reliable signal of fatigue. This design helps reduce false alarms that could occur during natural transitions or task switches.

**Fatigue Detection in Multi-Stage Tasks.** Many real-world tasks require different modalities at different stages. For instance, visual features may be important during image captioning but less relevant during an audio question-answering phase. Static fusion strategies often fail to adjust accordingly, leading to late or missing reactivation of needed modalities. Our system, however, tracks $\delta_m(t)$ and $\tau_m(t)$ separately for each modality. This allows it to detect local degradation even when other modalities are stable. As a result, interventions such as memory recall or weight adjustment can be triggered at the right time for the right stream.

**Interpretability and Local Signal Tracking.** Unlike attention-based methods that rely on large matrices or cross-modal maps, our approach models each modality using simple, one-dimensional signals. This makes the diagnosis process easier to interpret and visualize. Furthermore, since both $\delta_m(t)$ and $\tau_m(t)$ are computed from recent history without relying on future context, the system can operate in real-time settings. It does not require full-sequence access or gradient backpropagation, which makes it lightweight and deployment-friendly.

In conclusion, the interaction between $\delta_m(t)$ and $\tau_m(t)$ offers a precise, adaptive, and interpretable mechanism for detecting fatigue in each modality. These two signals together form the core of our MAD module. They support early detection of degradation, minimize unnecessary corrections, and make the system responsive to evolving task demands.

### A.3 GLOBAL ACTIVATION LANDSCAPE: DIAGNOSTIC PATTERNS

Modality fatigue often begins as local degradation, but its most pronounced effects emerge at the system level. Figure 3 captures this view by comparing the activation distributions $\{\alpha_m(t)\}$ across modalities in both healthy and fatigued states. In this section, we focus on two simple yet informative statistics: mean and variance, that help describe the global behavior of the system and provide diagnostic insight into reasoning failures.

**Mean Activation: Overall Responsiveness.** The average modality activation at time $t$, denoted $\mu_\alpha(t)$, measures how strongly the model is engaging with its available input streams. When the model is functioning well, $\mu_\alpha(t)$ tends to stay within a moderately high range, indicating that multiple modalities are actively contributing to the reasoning process. However, under fatigue, this mean value consistently declines. A low mean reflects a global drop in responsiveness, where the model fails to attend to any modality with sufficient strength. This condition frequently accompanies the symptom of *attention attenuation*, as described in Section A.1, where previously active modalities fade silently from the model's internal state. When $\mu_\alpha(t)$ remains low even though informative input is present, the result is often vague or unfocused predictions.

**Variance of Activation: Modality Specialization.** The second metric, variance of activation $\sigma_\alpha^2(t)$, reflects the system's ability to differentiate between modalities. High variance indicates that the model is attending selectively, amplifying relevant streams while suppressing less useful ones. This behavior is desirable in tasks where different modalities matter at different times. In contrast, a low variance implies that all modalities are treated similarly, regardless of their utility. This uniformity often signals a failure to reprioritize modalities and is characteristic of *task insensitivity*. In Figure 3, the fatigued state shows collapsed variance, where activation levels converge toward similar, low values. This convergence reduces the system's ability to adjust dynamically and weakens its overall reasoning capacity.

**Combinations of Mean and Variance: Diagnostic Patterns.**    By combining $\mu_\alpha(t)$ and $\sigma_\alpha^2(t)$, we can classify distinct failure modes based on their position in a two-dimensional diagnostic space:

- When both the mean and variance are low, the system is globally disengaged. This indicates that all modalities are underutilized and indistinguishable in importance. Such a pattern typically reflects the joint presence of *attention attenuation* and *task insensitivity*.

- A low mean combined with high variance suggests that only one modality is still active, while others have degraded. This imbalance is often linked to *semantic drift*, especially when the model defaults to text-based reasoning despite the availability of visual or auditory input.

- A moderate mean with increasing variance may signal *fusion instability*, where the model begins to over-rely on one modality to compensate for decline in another. This compensation leads to asymmetric attention allocation and unstable integration behavior.

These patterns show that simple global statistics can be highly informative. They not only confirm the presence of fatigue but also help interpret its form and severity.

**Real-Time Monitoring Implications.**    Both $\mu_\alpha(t)$ and $\sigma_\alpha^2(t)$ are lightweight to compute and do not depend on specific model architectures. This makes them practical for real-time diagnostic use. When the mean drops suddenly or the variance collapses, the system can flag potential fatigue and initiate interventions such as memory retrieval or fusion rebalancing. These global signals, when combined with per-modality traces like $\delta_m(t)$ from Section A.2, offer a complementary view. They help reveal systemic failures that may be overlooked by local signal analysis alone.

To summarize, the global activation landscape serves as an essential indicator of reasoning health in multimodal systems. While signals such as $\alpha_m(t)$ and $\delta_m(t)$ enable detailed monitoring at the modality level, the aggregated behavior reflected in $\mu_\alpha(t)$ and $\sigma_\alpha^2(t)$ exposes structural shifts in attention and integration. These two levels of analysis work together to explain, contextualize, and ultimately mitigate the behavioral symptoms described in Section A.1. As a result, our method offers both fine-grained detection and high-level observability, supporting robust control across diverse multimodal tasks.

# B    PSEUDOCODE–EQUATION CORRESPONDENCE

## B.1    MAPPING ALGORITHM 1 TO FORMAL EQUATIONS

To support transparency and reproducibility, we clarify how the high-level steps in Algorithm 1 correspond to formal equations presented in Section 4. Each operation in the pseudocode is implemented directly using one or more equations that define activation signals, fatigue detection, and fusion adjustments. Table 5 presents a structured mapping between algorithm lines and their mathematical definitions.

**Modular Implementation.**    The pseudocode structure follows a modular control loop. The first half (lines 1–5) focuses on signal extraction and fatigue diagnosis. These steps correspond to our MAD module, which monitors the health of each modality. The second half (lines 6–11) activates the MAC module to adaptively reweight and restore degraded modalities. The gating, recall, and fusion logic in these steps are directly instantiated by the equations in Section 4.2 and 4.3.

**Interpretability and Traceability.**    This mapping ensures that each symbolic operation in the algorithm has an interpretable and tractable implementation. Reviewers and practitioners can trace every design decision in the control process to a specific signal formula. As a result, the controller remains fully transparent, avoids heuristic shortcuts, and supports lightweight integration into existing multimodal systems.

In summary, Algorithm 1 serves as an abstract controller skeleton, where each step is precisely defined by a corresponding equation. This structure allows our system to be both formally grounded and computationally efficient, supporting real-time inference while maintaining interpretability.

Table 5: **Mapping between Algorithm 1 and key equations.**

| Algorithm 1 Step | Corresponding Equation(s) |
|---|---|
| Compute activation $\alpha_m(t)$ | Eq. (1): attention-weighted L2 norm of modality tokens |
| Compute attenuation rate $\delta_m(t)$ | Eq. (2): normalized activation change |
| Compute fatigue threshold $\tau_m(t)$ | Eq. (3): mean minus standard deviation over window |
| Determine fatigue flag $s_m(t)$ | Eq. (4): fatigue if $\delta_m(t) < -\tau_m(t)$ |
| Attenuate activation for fatigued $m$ | Eq. (6): decay $\alpha_m(t)$ using $(1 - \gamma)$ |
| Boost activation for non-fatigued $n$ | Eq. (6): enhance $\alpha_n(t)$ using relevance score $r_n(t)$ |
| Retrieve memory for fatigued modality | Eq. (7)–(8): attention-weighted sum of past features |
| Compute compensation gate $\lambda_m(t)$ | Eq. (9)–(10): similarity-based sigmoid function |
| Fuse memory and current input | Eq. (11): weighted average using $\lambda_m(t)$ |
| Normalize weights and compute output $z(t)$ | Eq. (12)–(13): softmax fusion over adjusted features |

### B.2 CONTROL PIPELINE RECAP

To support real-time intervention during modality fatigue, our controller is structured into a two-stage pipeline. The Modality Activation Decay Detector (MAD) handles signal extraction and early detection, while the Modality Alternation and Compensation Controller (MAC) adjusts fusion weights and restores weakened streams. Together, they implement a closed-loop control flow for detecting, responding to, and stabilizing multimodal degradation.

**Stage 1: Detecting Fatigue with MAD.** The MAD module continuously monitors each modality's internal state using activation level $\alpha_m(t)$ and change rate $\delta_m(t)$. Fatigue is detected when $\delta_m(t)$ drops below a dynamic threshold $\tau_m(t)$, triggering a binary fatigue signal $s_m(t)$:

$$s_m(t) = \begin{cases} 1 & \text{if } \delta_m(t) < -\tau_m(t) \\ 0 & \text{otherwise} \end{cases} \tag{15}$$

This signal flags modality $m$ as fatigued and passes control to the MAC module for corrective intervention.

**Stage 2: Adaptive Fusion with MAC.** Upon receiving the fatigue signal, MAC initiates a two-branch response:

- **For fatigued modalities** ($s_m(t) = 1$): MAC retrieves past memory vectors $M_m(t)$ using query-based attention and computes a compensation gate $\lambda_m(t)$ based on the similarity gap between the current and historical features. The corrected representation $\hat{f}_m(t)$ is then formed by blending the current input and memory:

$$\hat{f}_m(t) = \lambda_m(t) \cdot f_m(t) + (1 - \lambda_m(t)) \cdot M_m(t) \tag{16}$$

- **For healthy modalities** ($s_m(t) = 0$): MAC reweights their activation scores using a relevance metric $r_m(t)$ to preserve semantic clarity.

**Final Fusion and Output.** Once all modalities are updated, the controller computes a softmax-normalized attention distribution $w_m(t)$ using both the adjusted activation and relevance scores. The final output $z(t)$ is a weighted sum over all modality representations:

$$z(t) = \sum_m w_m(t) \cdot \hat{f}_m(t) \tag{17}$$

This formulation ensures that the final prediction incorporates both the preserved strength of healthy modalities and the restored signals from fatigued ones.

**Summary: Detect–Adapt–Restore Loop.** The control pipeline can be summarized as a sequential decision loop:

$$Monitor \rightarrow Trigger\ s_m(t) \rightarrow Recall\ M_m(t) \rightarrow Gate\ \lambda_m(t) \rightarrow Fuse\ z(t)$$

This loop reflects the controller's core design principle: track fine-grained changes in modality engagement and react locally with lightweight compensation. By tightly coupling detection and fusion, our system enables interpretable and modular correction of fatigue without disrupting the underlying model structure.

## C  DATASET SETUP AND TASK ORDER DETAILS

### C.1  DATASET STATISTICS AND MODALITIES

We evaluate our method on six multimodal reasoning benchmarks spanning static and temporal modalities. Table 6 summarizes dataset sizes, splits, and modality types.

Table 6: **Statistics of datasets used.**

| Dataset | Modality | #Samples | Split (Train/Val/Test) | Task Type |
|---|---|---|---|---|
| Flickr30K | Image + Text captions | 31,783 images, 158,915 captions | standard splits | Image captioning |
| AudioCaps | Audio + Text captions | approx. 46,000 audio clips | 49,838 / 495 / 975 | Audio captioning |
| MSR-VTT | Video + Text subtitles | 10,000 video clips, 200,000 captions | 9,000 / — / 1,000 | Video captioning |
| OK-VQA | Image + QA (knowledge) | approx. 14,000 QA pairs | standard split | Image QA |
| Clotho-AQA | Audio + QA pairs | 1,991 audio clips, 35,838 QA pairs | pre-defined split | Audio QA |
| MSVD-QA | Video + QA | approx. 50,000 QA pairs | pre-defined split | Video QA |

**Flickr30K** contains 31,783 images and 158,915 human-written captions (five per image). This dataset is widely used for benchmarking image-to-text captioning.

**AudioCaps** comprises approximately 46,000 audio clips, each with a single caption. The data is split into 49,838 training, 495 validation, and 975 test samples.

**MSR-VTT** includes 10,000 diverse video clips from the web. Each clip is annotated with 20 captions, totaling 200,000 clip-caption pairs. The standard evaluation split uses 9,000 training and 1,000 test videos.

**OK-VQA** is a knowledge-intensive image QA benchmark requiring external commonsense knowledge. The dataset contains about 14,000 QA pairs.

**Clotho-AQA** provides 1,991 audio clips. Each clip has 6 questions, each with 3 answers, resulting in a total of 35,838 QA pairs.

**MSVD-QA** is derived from MSVD video clips by generating QA pairs, forming a video-based QA task of similar scale to MSR-VTT QA.

### C.2  TASK ORDER DESIGN AND MOTIVATION

We define two fixed sequences of six tasks (Order A and Order B) designed to emphasize modality transitions and shifts in cognitive demand. These orders are crafted to increase the complexity of multimodal reasoning over time. Early tasks involve static and unimodal reasoning, while later tasks progressively introduce temporal dynamics, memory demand, and cross-modal inference. Frequent switches between modalities create natural fatigue scenarios.

### C.3  EVALUATION PROTOCOL AND SPLITS

All benchmarks follow their official data splits. We use training, validation, and test sets as provided by dataset authors. For round-wise evaluation of forgetting and recovery, only test splits are used across all rounds to prevent contamination of performance by overlap. Forgetting is quantified by re-evaluating prior task performance after new tasks have been introduced.

## C.4 ADDITIONAL SETUP DETAILS

**Preprocessing.** All image and video frames are resized to $224 \times 224$ resolution. Audio clips are resampled to 16 kHz and normalized. Text data (captions and questions) is tokenized using the tokenizer of the backbone language model.

**Modality Features.** For visual inputs, we use pretrained image (CLIP-ViT) and video encoders (TimeSformer). Audio features are derived from log-Mel spectrograms extracted using standard configurations.

**Task-specific Heads.** Captioning tasks are evaluated using CIDEr. QA tasks use accuracy as the main metric. All evaluation protocols follow prior standard benchmarks to enable fair comparison.

# D EVALUATION METRIC DEFINITIONS

We supplement our evaluation protocol by defining the fatigue detection and recovery behavior metrics used in Section 5.2. These metrics are aligned with the lifecycle modeling goals of our framework.

## D.1 FATIGUE DETECTION METRICS

**Fatigue Trigger Rate (FTR).** We compute the proportion of time steps where the fatigue signal $s_m(t) = 1$, representing activation-level decline for modality $m$. This metric reflects how frequently each modality enters a fatigue state across the evaluation window.

**Fusion-Bias.** Fusion bias quantifies the discrepancy between the actual fusion weight $w_m^{(t)}$ assigned to modality $m$ and its relevance score $r_m^{(t)}$ with respect to the query. Formally:

$$\text{FusionBias}_m^{(t)} = \left| w_m^{(t)} - r_m^{(t)} \right| \tag{18}$$

A high bias score indicates that the model either over-relied on or ignored a modality regardless of its contextual relevance, suggesting fatigue-induced misalignment.

**Entropy Change $\Delta$.** We track the change in attention entropy $H_m^{(t)}$ of modality $m$ to measure attention dispersion over time. The entropy is defined as:

$$H_m^{(t)} = -\sum_{i=1}^{N_m} a_i^{(t)} \cdot \log\left(a_i^{(t)}\right) \tag{19}$$

where $a_i^{(t)}$ is the attention score of the $i$-th token in modality $m$ at time $t$. An increase in $H_m^{(t)}$ indicates broader attention dispersion, which may be an early signal of fatigue.

**Average Forgetting.** To evaluate performance degradation due to fatigue, we compute the average accuracy drop across task stages compared to the initial stage. This metric captures how well the model preserves cross-task consistency as modalities become fatigued.

## D.2 RECOVERY BEHAVIOR METRICS

**Recovery Gain.** We define recovery gain as the performance improvement immediately after a fatigue trigger ($s_m(t) = 1$) compared to the preceding step. This reflects the model's ability to utilize the memory path to recover lost modality fidelity.

**Compensation Usage.** This metric records the frequency at which the memory path is invoked after a fatigue trigger, indicating how often the model relies on reactivation mechanisms to restore modality performance.

**Supporting Distribution Visualizations.** To provide interpretability, we plot the temporal distribution of $s_m(t)$ triggers, fusion weights $w_m(t)$, and entropy $H_m(t)$ across modalities. Visual indicators such as moving averages or sample window overlays are used to facilitate comparisons between healthy and fatigued states.

### D.3 Auxiliary Faithfulness Diagnostic (Optional)

In supplementary analysis, we measure the semantic faithfulness of outputs to the current modality:

$$\text{Faith}_m^{(t)} = \text{sim}\left(\text{Answer}^{(t)}, \text{Modality}_m^{(t)}\right) \tag{20}$$

where $\texttt{Answer}^{(t)}$ is the textual output at step $t$, and $\texttt{Modality}^{(t)}$ refers to the full semantic embedding of the current modality (e.g., CLIP image vector, audio embedding). A low similarity score implies that the response has drifted from visual/audio grounding, a potential symptom of semantic hallucination induced by modality fatigue.

## E Full Breakdown of Task-Level Metrics

We provide a detailed breakdown of the metrics reported in Table 1, focusing on two aspects: (1) task-level forgetting traces under different task orders, and (2) modality-specific degradation patterns across diverse task types. While no additional plots are included, we offer precise textual descriptions ofW the metric behaviors to guide interpretation and future replication.

### E.1 Per-Task Forgetting Traces

We decompose the aggregated *Forgetting Ratio* (FR) into per-task components across both task sequences (Order A and Order B). This enables a granular analysis of how different tasks contribute to the overall forgetting trend.

- **Temporal Trends.** For each task, we trace the dynamic evolution of the modality activation $\alpha_m(t)$ over multiple rounds. Tasks such as *MSVD-QA* and *OK-VQA* exhibit sustained declines in $\alpha_m(t)$ across modalities, with frequent fatigue triggers $s_m(t) = 1$ indicating accumulating strain.

- **Order Sensitivity.** When comparing Orders A and B, we find that certain tasks show earlier onset of fatigue in one order versus the other. For example, *Image Captioning* demonstrates more stable retention when presented earlier in Order A, but decays faster in later rounds of Order B.

- **Task Difficulty.** Tasks requiring fine-grained multimodal reasoning (e.g., open-ended QA or cross-modal captioning) tend to show higher per-task FR, confirming their vulnerability to fatigue-induced forgetting.

### E.2 Modality-Specific Forgetting Patterns

Beyond task decomposition, we further examine how different modalities degrade across the six evaluation tasks, highlighting non-uniform vulnerability.

- **Visual modality** shows the highest degree of activation decay in tasks like *Image Captioning* and *OK-VQA*, reflecting the cognitive cost of fine-grained image-text alignment. Fatigue events are clustered near visual reasoning prompts, often following prolonged exposure to static visual scenes.

- **Auditory modality** demonstrates comparatively slower degradation in *Audio Captioning* and *Clotho-AQA*, though it becomes more fragile under ambiguous or cross-modal references (e.g., audio-related reasoning in multi-turn QA).

- **Video modality** exhibits periodic activation drops in both *MSR-VTT* and *MSVD-QA*, consistent with its high temporal processing demands. Memory decay for video is more bursty, with steep losses following clips containing frequent scene transitions.

Taken together, these observations confirm that modality fatigue is not uniform: different modalities decay at different rates depending on the task context, temporal position, and cognitive demand. This underlines the necessity of adaptive memory regulation and modality-aware scheduling in long-horizon multimodal systems.

## F  REPRODUCIBILITY

### F.1  HARDWARE AND SOFTWARE ENVIRONMENT

All experiments were conducted on a single NVIDIA RTX A6000 GPU (48 GB memory). We implemented our methods in the PyTorch framework and rebuilt the multimodal pipeline using the HuggingFace Transformers interface. During all experiments, both the multimodal encoders and the language backbone remained frozen; only the inserted modules were optimized to ensure controlled resource overhead.

### F.2  MULTIMODAL MODEL CONFIGURATION

Every method shares the same multimodal encoder structure and language generation backbone. The language model backbone is LLaMA-3.2B-Instruct, fine-tuned in a lightweight fashion via LoRA, optimizing only the adapter weights. The image and video encoder is the pretrained EVA-CLIP-B (ViT-g backbone), and the audio encoder is the pretrained BEATs model; these remain frozen throughout. Each modality uses its own projection layer: the image modality projection consists of two MLP layers, while the audio and video modalities employ a lightweight convolutional plus MLP structure. A unified cross-modal fusion module in the form of cross-attention is integrated, and all baselines deploy on this common framework. The MAD and MAC methods insert external control modules for modality fatigue diagnosis and compensation, without altering the backbone architecture.

### F.3  BATCH SIZE AND PRECISION SETTINGS

Training is performed separately for each single-modality task without mixing modalities. The default batch size is set to 32, with automatic mixed-precision (fp16) enabled to improve memory efficiency. In memory-constrained scenarios, gradient accumulation with two steps is supported. During inference, we use a batch size of 16 under `torch.no_grad()` to ensure stable evaluation.

### F.4  MAD AND MAC IMPLEMENTATION DETAILS

The MAD (Modality-Aware Drift) module detects task fatigue or distribution shifts in the current input modality via a gating mechanism that generates corrective vectors. The MAC (Modality Alternation & Compensation) module injects historical fused features to guide compensation decisions once an anomaly is detected. Both modules are plugged into the cross-modal attention layers before text generation, without structural changes. Each consists of several shallow linear layers; the total additional parameters remain under one million, and inference overhead does not exceed 3.5%.

### F.5  TRAINING AND RESOURCE CONSUMPTION

All encoder parameters remain frozen, and only MAD and MAC module parameters are optimized. We use the AdamW optimizer with an initial learning rate of $1\times10^{-4}$, a linear decay schedule, and weight decay of $1\times10^{-2}$. No replay buffers or auxiliary generators are employed. The total training cost is approximately 120 GPU-hours, substantially lower than full-model fine-tuning, making the approach well suited for efficient continual-learning deployment.

### F.6  EVALUATION SETTINGS

We evaluate on six task–modality combinations covering two task types (Captioning and QA) across three modalities (Image, Audio, Video). Captioning tasks use Flickr30k for images, AudioCaps for audio, and MSR-VTT for video; QA tasks use OK-VQA for images, Clotho-AQA for audio, and MSVD-QA for video. Each evaluation round is conducted on the task's test set with all parameters

frozen. Standard metrics (CIDEr for captioning, Accuracy for QA) are reported alongside four fatigue indicators: Forgetting Rate, Fusion Bias, Entropy, and Faithfulness Drop. Results are averaged over three random seeds, with standard deviations reported for stability.

## LLM USAGE DISCLOSURE

In accordance with the ICLR 2026 policy on large language model (LLM) usage, we disclose that LLM assistance (OpenAI GPT-5 via ChatGPT) was employed during the preparation of this paper. Specifically, the LLM was used for (i) refining the clarity and readability of text, (ii) restructuring sections for better logical flow, and (iii) generating illustrative figure captions and LaTeX formatting templates. All technical content, including problem formulation, theoretical derivations, experimental design, and result interpretation, was conceived, implemented, and validated solely by the authors. The LLM did not contribute to the novelty of the research ideas, data collection, analysis, or conclusions.

