# OpenReview forum: "When Multimodal Models “Burn Out”: Diagnosing and Healing Modality Fatigue via MAD + MAC"
_ICLR.cc/2026/Conference — ICLR 2026 Conference Withdrawn Submission_

### Official Review · Reviewer_ZKfS · 2025-10-31

**Soundness:** 3
**Presentation:** 3
**Contribution:** 3
**Rating:** 6
**Confidence:** 1

**Summary:**

This paper introduces the concept of modality fatigue, arguing that multimodal LLMs gradually stop using certain modalities during long-horizon, multi-step inference. The paper proposes a two-part controller MAD and MAC that forms a detect-adapt-restore feedback loop and empirically reduces catastrophic forgetting and fusion bias.

**Strengths:**

1. The paper identifies and formalizes modality fatigue, giving an interpretable, measurable account of how specific modalities silently fade during long, multi-modal reasoning instead of treating fusion failure as a one-step attention issue.
2. The proposed MAD+MAC loop is conceptually clean and practically appealing.
3. The experimental setup explicitly continual, cross-task inference and reports forgetting over time, which is much closer to real assistant-style usage than static single-task benchmarks.

**Weaknesses:**

See in questions.

**Questions:**

1. Could the fatigue detector be falsely flagging a modality as “fatigued” in cases where that modality is simply no longer needed at that step, and do you have any measurement of such false positives?
2. When you restore a fatigued modality by injecting its past representations, how do you prevent reintroducing stale or incorrect information from earlier steps and amplifying that error?
3. What is the actual inference-time overhead (latency, memory, throughput) of MAD+MAC, and can the method still run in long-horizon or near–real-time settings without becoming too slow or heavy?

---

### Official Review · Reviewer_dRn7 · 2025-10-31

**Soundness:** 2
**Presentation:** 2
**Contribution:** 2
**Rating:** 2
**Confidence:** 4

**Summary:**

This work attempts to contextualize modality collapse in long context multimodal reasoning through defining a phenomenon termed “Modality Collapse”. The paper argues that long context reasoning in multimodal models fails owing to “overall reduction in activity level of a modality” as the generation length increases, resulting in faulty reasoning and answers. Through modelling this attention collapse as a continuous state variable, a fatigue detector (MAD) is defined along with a modality compensator (MAC) in order to improve reasoning capabilities of multimodal models. The effectiveness of the proposed method is validated through evaluation on a combination of audio, image and video evaluation datasets (AudioCaps, MSR-VTT, OK-VQA and others.)

**Strengths:**

The current work attempts to propose an interesting concept of attempting to identify timesteps during multimodal reasoning wherein the model starts to decay importance to video/audio modality, along with attempting to compensate for this identified decay. This idea is interesting in the MLLM domain in order to address modality collapse.

**Weaknesses:**

The authors define the central concepts such as modality fatigue and provide appropriate definitions, but the present work is not contextualized in the midst of contemporary works on modality collapse. Additionally, the evaluation is not setup well. Also, the authors do not use multimodal models, rather train their own fusion model. The details of which are not clear and it is not clear whether the authors perform image, audio, video modality alignment, SFT and Preference tuning before the experiments.

While the paper claims to be addressing multimodal long-context reasoning, the authors use a combination of ImageQA, VideoQA, AudioQA datasets in order to perform the long-context reasoning, through multiple questions in a single conversation with the MLLM. However, this cannot be considered as long-context as the MLLM would not require much of a context in order to answer the respective question. It would require to attend to the current image/audio/video in question.

During the definition of modality fatigue, the manuscript does not provide validation of whether the multimodal models tend to provide faulty reasoning when the fatigue threshold is hit. Providing this experiment could add validation to the proposed threshold of modality fatigue.

In the current work, the defined phenomenon of modality fatigue is not empirically shown in current MLLM (open source models such as QwenVL2.5, Phi-4, Qwen 3 omni). The trained MLLM model has a small training dataset, which does not perform the appropriate scale of SFT followed by Instruction tuning in order for the model to follow the multiple rounds of the evaluation task. Hence, the stated results may or may not translate to MLLM models.

**Questions:**

In Section 3.2, the assumption is that MACC is active only when s_mt=1. Hence when s_mt=1, Eq.6 would simplify to (alpha~_mt = alpha_mt - (1-gamma)alpha_mt). It is not clear from the manuscript what the constant gamma is? Also, this does not use r_mt defined in Eq.5. Additionally, how is f_mt defined? Is it the average of all the embeddings of modality m?

In Fig. 2a, it is not very clear whether the plots are an average across different questions, which dataset the particular example was taken from?

The rationale behind choosing activation level of modality m to be the attention weighted sum of L2 norm of hidden states is unclear. It is stated that “average attention” would miss per modality variations. However, no validation/appropriate citations are provided in support of this claim.

Line 19: Could the authors clarify the term “equationment graphs”, is it a typo?

In order to test long-context reasoning, authors should consider employing long context benchmarks such as VideoMME: https://arxiv.org/pdf/2405.21075 (contains video understanding questions with video durations ranging from a few minutes to ~60 minutes) Long Video Bench: https://arxiv.org/pdf/2407.15754 or any other datasets wherein the questions require the model to parse through a long-context input.

---

### Official Review · Reviewer_769R · 2025-10-31

**Soundness:** 3
**Presentation:** 3
**Contribution:** 3
**Rating:** 4
**Confidence:** 4

**Summary:**

This paper studies the scenario when multimodal models, when dealing with long-context inputs, tend to ignore some of the modalities, something that the authors call modality fatigue (and is also known as modality collapse in existing literature). The authors propose a mechanistic way of monitoring the representativeness of a modality within a model, which they use as a signal to detect when a modality might be undergoing collapse and intervene accordingly based on reweighting and recall of past information. They empirically evaluate their method on several multimodal language modelling tasks accompanied by serval analytical experiments.

**Strengths:**

1. The authors propose a mechanistic way of monitoring the relative contributions of modalities dynamically with increasing context lengths in multimodal language modelling tasks, which is a novel contribution. This allows for an empirically principled tackling of modality collapse with behaviours of the model that can be monitored at the low level and not just the performance on the end-task.

2. The idea of reweighting / reincorporating modalities undergoing collapse based on prior memory in long-context learning is an interesting idea and seems to be empirically effective.

3. The paper is well written and the core concepts and propositions are clearly and intuitively explained.

4. Empirical evaluation is sufficient, where the authors showcase not only the effectiveness of their interventions on the downstream task performance, but how their approach of monitoring modality health actually correlates with real model behaviour in long-context multimodal learning.

**Weaknesses:**

1. The paper misses the entire existing literature on modality collapse, which is essentially the same issue as what is described here as modality fatigue, and is something that is being studied actively in the multimodal learning community with several recent developments [a, b, c].

2. For an overall system level indicator of activation imbalance, just the mean and variance of $\alpha_m(t)$ may not be sufficient for monitoring activation imbalance, since shifting means can correspond to changing activation patterns and both high (some activations are very high and others very low) and low variance can signify imbalance, i.e., the arguments presented in Line 196-200 may not necessarily hold. The entropy of $\alpha_m(t)$ seems like a better indicator of activation imbalance.

3. Prior literature such as [c] shows that the probability that a multimodal model would ignore certain modalities increases with the number of modalities. Along these lines, based on the proposed notion of modality fatigue, it can be expected that the probability of fatigue would also increase as the number of modalities increase. It would be interesting to know whether this actually happens, which can also help establish the relationship of this work with the existing literature.

4. Although the authors argue that the elimination of modalities can be disadvantageous, they do not take into account the intrinsic importance of modalities while making this argument. Some modalities are inherently more informative than others, or at least, are easier to learn than others; for instance, as, pointed out in [d] different modalities can have different convergence rates. In some other cases, focussing on just one could provide better robustness than focussing on many [e]. Thus, manually reweighting modalities, resulting in a shift in the natural weights assigned as part of gradient descent, could harm downstream task performance in these scenarios.

5. By extension to the above point, the following question could be asked: what if the collapse of a modality is a necessary action, for instance, when the modality is noisy? In that case, forcing the model to include it could negatively impact its performance. Is there a way to tell apart when the model is reasonably eliminating signals from a modality vs when it is suffering from fatigue? Can this information somehow be inferred from the proposed descriptors of modality health and leveraged to tackle this situation?

Minors:\
Line 13: Use ``(double backticks) for opening double quotes in LaTeX.\
Line 21: "vital sign," -> "vital sign",\
Line 16: "They generalizes" -> "They generalize"

References:\
[a] Javaloy et al., "Mitigating modality collapse in multimodal VAEs via impartial optimization", ICML 2022.\
[b] Wu et al., "Multimodal patient representation learning with missing modalities and labels", ICLR 2024.\
[c] Chaudhuri et al., "A Closer Look at Multimodal Representation Collapse", ICML 2025.\
[d] Wang et al. What makes training multi-modal classification networks hard? CVPR 2020.\
[e] Liu et al., "Comparison of the Robustness of Multimodal Models and Unimodal Models under Text-based Adversarial Attacks", International Conference on Machine Learning and Automation 2024.

**Questions:**

Please refer to the Weaknesses section.

---

### Official Review · Reviewer_khBd · 2025-11-01

**Soundness:** 2
**Presentation:** 2
**Contribution:** 2
**Rating:** 2
**Confidence:** 4

**Summary:**

This paper proposes a new perspective on the phenomenon of "modality fatigue" in long-context multimodal reasoning, where certain input modalities progressively lose their influence over time during reasoning. To address this, they propose a framework that detects modality fatigue in real time using a Modality Activation Decay Detector (MAD), and compensates using a Modality Alternation & Compensation Controller (MAC). The proposed approach aims to maintain modality balance and enhance reasoning stability, without requiring significant architectural modifications.

**Strengths:**

- The idea of modality fatigue is interesting. The approach provides a new perspective for long-context reasoning.
- The MAD and MAC modules can be integrated into existing MLLMs without requiring drastic changes to the architecture, making it accessible.

**Weaknesses:**

- While the authors present analysis of modality fatigue, there is limited theoretical or empirical justification for this phenomenon. More actual evidence is absolutely needed.
- The authors argue that "burn out" happens when input length increases and reasoning deepens. However, the experimental settings seem more aligned with continual learning rather than the long-context reasoning scenario described. The experimental settings are not well aligned with the analysis.
- Figure 4, which outlines the methodology, is not very clear in terms of its illustration. The authors could improve the clarity of this figure, especially in showing how the detection and compensation mechanism works.
- In Equation 5, the term $f_m$ is not defined, making it confusing.
- The paper lacks evaluation on more general benchmarks, like MME. It is not clear whether the proposed method can enhance MLLMs or not.
- There is insufficient detail about the specific MLLMs used in the experiments. Providing more information about these settings would be helpful.

**Questions:**

Please check the above section.

---

### Note · Authors · 2025-11-12

I have read and agree with the venue's withdrawal policy on behalf of myself and my co-authors.